# Hydrolysis of Oligodeoxyribonucleotides on the Microarray Surface and in Solution by Catalytic Anti-DNA Antibodies in Systemic Lupus Erythematosus

Tatiana S. Novikova [1,2], Evgeny A. Ermakov [1,2,*,†], Elena V. Kostina [1,†],
Alexander N. Sinyakov [1], Alexey E. Sizikov [1,3], Georgy A. Nevinsky [1,2] and Valentina N. Buneva [1,2,*]

[1] Institute of Chemical Biology and Fundamental Medicine, Siberian Branch of the Russian Academy of Sciences, 630090 Novosibirsk, Russia
[2] Department of Natural Sciences, Novosibirsk State University, 630090 Novosibirsk, Russia
[3] Institute of Clinical Immunology, Siberian Branch of the Russian Academy of Sciences, 630099 Novosibirsk, Russia
* Correspondence: evgeny_ermakov@mail.ru (E.A.E.); buneva@niboch.nsc.ru (V.N.B.)
† These authors contributed equally to this work.

**Abstract:** Anti-DNA antibodies are known to be classical serological hallmarks of systemic lupus erythematosus (SLE). In addition to high-affinity antibodies, the autoantibody pool also contains natural catalytic anti-DNA antibodies that recognize and hydrolyze DNA. However, the specificity of such antibodies is uncertain. In addition, DNA binding to a surface such as the cell membrane, can also affect its recognition by antibodies. Here, we analyzed the hydrolysis of short oligodeoxyribonucleotides (ODNs) immobilized on the microarray surface and in solution by catalytic anti-DNA antibodies from SLE patients. It has been shown that IgG antibodies from SLE patients hydrolyze ODNs more effectively both in solution and on the surface, compared to IgG from healthy individuals. The data obtained indicate a more efficient hydrolysis of ODNs in solution than immobilized ODNs on the surface. In addition, differences in the specificity of recognition and hydrolysis of certain ODNs by anti-DNA antibodies were revealed, indicating the formation of autoantibodies to specific DNA motifs in SLE. The data obtained expand our understanding of the role of anti-DNA antibodies in SLE. Differences in the recognition and hydrolysis of surface-tethered and dissolved ODNs need to be considered in DNA microarray applications.

**Keywords:** systemic lupus erythematosus; SLE; anti-DNA antibodies; antinuclear antibodies; natural catalytic antibodies; abzymes; recognition; sequence specificity; microarray; DNase I





## 1. Introduction

Systemic lupus erythematosus (SLE) is an autoimmune disease characterized by heterogeneous clinical manifestations and the production of a plethora of autoantibodies that form immune complexes and mediate tissue damage [1,2]. Antinuclear autoantibodies, including anti-DNA antibodies, are known to be serological hallmarks of SLE [3,4]. Anti-DNA antibody assays have demonstrated clinical utility for diagnosing SLE and monitoring disease activity [3–5]. According to the recommendations of the European League Against Rheumatism (EULAR) and the American College of Rheumatology (ACR), the presence of anti-DNA antibodies is one of the classification criteria for SLE [6]. However, anti-DNA antibody levels vary with the course of SLE, sometimes decreasing to undetectable values [7,8]. Therefore, only 30–50% of patients are anti-DNA antibody positive at some point during the course of the disease [5,8].

DNA complexed with proteins serves as an antigen for the generation of anti-DNA antibodies, but its origin remains poorly understood. There are two hypotheses about the origin of DNA as the initial antigen in SLE [9]. According to the first hypothesis,

self-DNA released in the form of DNA complexes with nucleosomes or DNA-binding proteins during cell death is an antigen in SLE [4,10,11]. Self-DNA recognition through Toll-like receptors (in particular, TLR7 and TLR9) and the initiation of the inflammatory response also contribute to the breakdown of immunological tolerance and the generation of anti-DNA antibodies in SLE [11]. Data on increased levels of cell-free DNA in the plasma of SLE patients compared with healthy individuals, support this hypothesis [12,13]. Thus, cell death by apoptosis, necrosis, or NETosis are assumed to be the most common source of autoantigens in SLE [3,4,14]. However, according to the second hypothesis, bacterial DNA can also become an antigen for the production of anti-DNA antibodies, given its immunostimulating activity [9]. The presence of unmethylated CpGs and other structural motifs in bacterial DNA, in contrast to mammalian DNA, greatly increases its immunogenic properties [15]. Therefore, anti-DNA antibodies in SLE may recognize both bacterial and self-DNA.

Data on the specificity of DNA recognition by autoantibodies are quite limited. Available data indicate that anti-DNA antibodies recognize single-stranded (ssDNA), double-stranded (dsDNA), and specific forms of DNA (e.g., B-DNA and Z-DNA) [16,17]. Electrostatic interactions with the DNA backbone are thought to play an important role in DNA binding by antibodies [18–20]. The enrichment of positively charged amino acids, such as arginine, in the complementarity determining regions (CDRs) of anti-DNA antibodies, promote interactions with the negatively charged phosphate groups of the DNA backbone [19,21]. However, non-electrostatic interactions may also occur [20]. Nucleotide sequence and secondary nucleic acid structure are also important for binding, as evidenced by data on the specificity of anti-DNA antibodies to both conserved and non-conserved sequences [18,22,23]. Nonetheless, the sequence specificity of anti-DNA antibodies is still poorly understood.

The spectrum of anti-DNA antibodies in SLE is represented by various isotypes. IgG antibodies are considered the most significant isotype causing pathogenic reactions in SLE [4]. However, anti-DNA IgM antibodies, called natural autoantibodies, are also found in SLE [24]. Natural antibodies bind foreign and self-antigens and represent the first line of host defense against pathogens [25,26]. These antibodies may belong to the IgM, IgG, or IgA classes and are thought to be involved in the clearance of apoptotic debris, including DNA-protein complexes. Natural antibodies exhibit lower affinity and cross-reactivity compared to antigen-specific antibodies [26]. Among natural antibodies, IgGs with catalytic properties called abzymes, are also known [27–29]. Interestingly, catalytic antibodies that recognize and hydrolyze DNA were first discovered in SLE patients [30]. Moreover, using a number of criteria, it was proven that DNase activity was caused by antibodies, and not by any hypothetical impurities of other proteins. Monoclonal catalytic antibodies exhibiting DNase activity were also obtained [31,32]. Data on the catalytic activity of DNA-hydrolyzing abzymes in SLE, multiple sclerosis, and several other autoimmune and viral diseases have been summarized in several reviews [27–29,33,34]. However, the sequence specificity of anti-DNA IgG with catalytic activity in SLE has not been studied.

Several methods for analyzing antibody–antigen (or enzyme–substrate) interactions have shown differences in protein recognition of antigen/substrate in solution or when immobilized on a surface [35–37]. Immobilization of antigen/substrate on the surface of microarrays allows for the development of high-throughput systems [38]. Natural catalytic anti-DNA antibodies have lower affinity for antigen compared to antigen-specific antibodies [28], so traditional analytical methods (for example, enzyme-linked immunosorbent assay, ELISA) are not suitable for in-depth evaluations of short DNA recognition. In addition, their catalytic properties also need to be considered. Therefore, in this work, we analyzed the hydrolysis of short DNA, immobilized on microarray surfaces and in solution, by catalytic anti-DNA antibodies from SLE patients.

## 2. Materials and Methods

### 2.1. Study Participants and Biological Material

This study was approved by the Local Ethics Committee of the Institute of Chemical Biology and Fundamental Medicine (ethical approval protocol N3 from 19 June 2023).

Five patients with active SLE and five healthy individuals were recruited for this study at the Institute of Clinical Immunology (Novosibirsk, Russia). The inclusion criteria for patients were as follows: diagnosis of SLE (M32, ICD-10) in accordance with the Russian Association of Rheumatologists and EULAR recommendations [6,39], active phase of the disease, anti-dsDNA and anti-ssDNA antibody positivity, over 18 years of age, and signed consent. Vecto-dsDNA-IgG (Cat. # 8656) and Vecto-ssDNA-IgG (Cat. # 8658) ELISA kits (Vector-BEST, Novosibirsk, Russia) were used to detect anti-DNA antibodies. A level of >25 IU/mL for anti-dsDNA and >25 U/mL for anti-ssDNA was considered positive. The SELENA-SLEDAI (Safety of Estrogens in Lupus Erythematosus National Assessment–SLE Disease Activity Index) scale was applied to assess SLE disease activity [40]. Each patient received at least two types of drugs affecting the immune system, one of which belonged to the corticosteroid class (dexamethasone, prednisolone, methylprednisolone, or betamethasone). The dosage of corticosteroid ranged from 1 to 20 mg/day, depending on the drug (as in [41]). Other drugs included methotrexate, hydroxychloroquine, celecoxib, azathioprine, filgrastim, tenoxicam, and mycophenolate mofetil. Exclusion criteria for patients were autoimmune diseases (except SLE), recent infectious diseases, cancer, or other concomitant somatic diseases in the acute stage. Healthy individuals without active somatic pathology were recruited as a comparison group. Serum obtained according to the method described earlier [41] was used as a biological material for analysis.

### 2.2. Purification of Antibodies from the Serum

Antibodies (IgG) were purified from the serum using the affinity chromatography method described previously [42,43]. Briefly, serum was centrifuged for 5 min at 10,000 rpm and then diluted 1:3 with Tris-Buffered Saline (TBS: 50 mM Tris-HCl pH 7.5 and 150 mM NaCl). The diluted serum was applied to a HiTrap Protein G HP antibody purification column (1 mL) (Cytiva, Uppsala, Sweden) using an ÄKTA Start chromatography system (Cytiva, Uppsala, Sweden). The column was washed with TBS and TBST (TBS containing 1% Triton X-100), and then the proteins were eluted with 100 mM Gly-HCl (pH 2.6). IgG samples were further purified by FPLC gel filtration on a Superdex 200 HR 10/30 column (Cytiva, Uppsala, Sweden). The resulting IgG fractions were immediately neutralized with 1.0 M Tris-HCl buffer (pH 8.8) and then microdialyzed in 50 mM Tris-HCl (pH 7.5). Fractions from the central part of the chromatogram were used for subsequent analysis. The IgG concentration was determined by spectrophotometry. The purity and electrophoretic homogeneity of IgG samples was confirmed by electrophoresis in a 4–15% polyacrylamide gel followed by staining with Coomassie G-250, similar to [42,43].

### 2.3. Model Oligodeoxyribonucleotides Used for Analysis

In this work, thirteen model oligodeoxyribonucleotides (ODNs) synthesized at the ICBFM SB RAS on an automatic eight-column synthesizer ASM-800 (Biosset LLC, Novosibirsk, Russia) using standard methods were used as substrates for the analysis of nuclease activity. The names and sequences of the ODNs are presented in Table 1. The indicated ODNs were selected based on: (1) nucleotide type (homoODNs); (2) sequence (ODNs with alternating nucleotides); (3) available data on specific single-stranded DNA motifs recognized by anti-DNA antibodies [22].

**Table 1.** Model oligodeoxyribonucleotides (ODNs) used in this study.

| ODN Name | Sequence | Number of Nucleotides | Sequence Description |
|---|---|---|---|
| $A_{10}$ | ROX-5′-AAAAAAAAAA [1] | 10 | consists entirely of adenosines |
| $C_{10}$ | ROX-5′-CCCCCCCCCC [1] | 10 | consists entirely of cytidines |
| $T_{10}$ | ROX-5′-TTTTTTTTTT [1] | 10 | consists entirely of thymidines |
| $(AC)_5$ | ROX-5′-ACACACACAC [1] | 10 | consists of alternating purine and pyrimidine nucleobases |
| $(AT)_5$ | ROX-5′-ATATATATAT | 10 | consists of alternating purine and pyrimidine nucleobases |
| $(GT)_5$ | ROX-5′-GTGTGTGTGT [1] | 10 | consists of alternating purine and pyrimidine nucleobases |
| $(AG)_5$ | ROX-5′-AGAGAGAGAG [1] | 10 | composed of alternating purine bases |
| $(CT)_5$ | ROX-5′-CTCTCTCTCT [1] | 10 | consists of alternating pyrimidine bases |
| $(CT)_3$ | ROX-5′-CTCTCT-linker-NH2 [2] | 6 | ODNs consisting of alternating pyrimidine bases varying in length |
| $(CT)_5$ | ROX-5′-CTCTCTCTCT-linker-NH2 [2] | 10 | |
| $(GT)_3$ | ROX-5′-GTGTGT-linker-NH2 [2] | 6 | ODNs consisting of alternating purine and pyrimidine nucleobases varying in length |
| $(GT)_5$ | ROX-5′-GTGTGTGTGT-linker-NH2 [2] | 10 | |
| $C_{14}T_{10}$ | ROX-5′-CCCCCCCCCCCCCCTTTTTTTTTT-linker-NH2 [2] | 24 | Long ODN consisting of 24 nucleotides |

[1] ODNs similar in sequence but containing a linker and 3′-terminal NH2 were used in microarray experiments.
[2] Linker = aminohexyl linker + 6 triethylene glycol phosphate residues.

The ODNs were fluorescently labeled with 5-carboxy-rhodamine-X (ROX) at the 5′ end. Oligonucleotides for microarray experiments additionally contained an aminohexyl linker at the 3′ end (for attachment to the surface), and 6 triethyleneglycol phosphate residues to extend the linker part. $(AT)_5$ was not used for microchip studies due to limited surface space.

### 2.4. Microarray Slide Production

Microarray were printed on glass epoxy slides using the contact printing method on a BioOdyssey Calligrapher Miniarrayer spotter (Bio-Rad, Feldkirchen, Germany). Each slide contained 12 wells with microarrays of 6 × 6 or 6 × 8 spot probes suitable for simultaneous analysis. The average diameter of the spots (microarray cells with immobilized probes) was ~360 µm, and the distance between the spots was 750 µm. Printing was carried out in 135 mM $NaHCO_3$. The concentration of oligonucleotides was 2 µM. Each oligonucleotide was printed in 3–6 replicates on each microarray. After printing, the slide was washed sequentially in $H_2O$, 0.05% sodium dodecyl sulfate (SDS), $H_2O$, 50 mM Tris-HCl pH 7.5, and centrifuged.

The slide was scanned on a ScanArray Express 2.0 scanner (Perkin Elmer, Rodgau, Germany) using an exciting laser with a wavelength $\lambda_1 = 540$ nm. Image analysis was carried out using the Scanarray Express program (Perkin Elmer, Rodgau, Germany).

### 2.5. Analysis of Hydrolysis of Immobilized ODNs on Microarray Surface by Antibodies

The analysis of ODN hydrolysis by IgG antibodies or the control enzyme deoxyribonuclease I (DNase I) (Cat #: M0303S, New England Biolabs, Ipswich, MA, USA) was carried out in a reaction mixture (70 µL) containing 50 mM Tris-HCl pH 7.5, 5.0 mM $MgCl_2$, 0.1 mg/mL (0.67 µM) IgG. For DNase I, the reaction mixture consisted of 1X buffer supplied with the enzyme and 0.02 U/µL DNase I. The resulting mixture was applied to a microarray well and incubated in a thermostat at 37 °C from 30 min to 4 h. The buffer solution was applied to the control well. After incubation, the slides were washed sequentially in $H_2O$, 0.05% SDS, $H_2O$, 50 mM Tris-HCl pH 7.5, centrifuged, and scanned. To reduce possible

slide-to-slide variation in ROX fluorescence, the data obtained were normalized to the fluorescence value for 3′-ROX mononucleotide T (which cannot be hydrolyzed) printed in triplicate on each microarray. Data from three to six replicates were averaged and normalized for each ODN. Experiments were carried out in 1–3 replicates.

*2.6. Analysis of ODN Hydrolysis in Solution by Antibodies Using Electrophoresis*

A reaction mixture (10 μL) containing 50 mM Tris-HCl pH 7.5, 5 mM MgCl$_2$, 5.7 μM of one of the ODNs, and 0.1 mg/mL (0.67 μM) IgG was prepared. The control mixture did not contain antibodies. The reaction mixture was incubated at 37 °C for 15–30 min, depending on the activity of IgG samples (hydrolysis efficiency). The final reaction time was 15 min for SLE33 and SLE22, and 30 min for SLE44, SLE23, and SLE24. Incubation time for IgG samples of healthy donors was 30 min. For DNase I, the reaction mixture consisted of 1X supplied buffer and 0.1 U/μL DNase I, and incubation time was 30 min. All experiments were performed in two independent replicates. After incubation, 10 μL of denaturing buffer (8 M urea and 0.025% xylene cyanol) was added to stop the reaction.

The reaction products were detected using denaturing gel electrophoresis In a 20% polyacrylamide gel consisting of 2.81 M Acrylamide (AA), 72.6 mM bis-AA, 7 M urea, 89.2 mM Tris, 89 mM H$_3$BO$_3$, and 2.0 mM EDTA, similar to [44]. Electrophoresis was carried out for 1 h 45 min at 800 V and 50 mA using the Power supply ELF-8 (DNA-Technology LLC, Moscow, Russia). The results of gel electrophoresis were recorded using an Amersham Typhoon laser scanner (Cytiva, Uppsala, Sweden). To determine the relative level of hydrolysis (%), the intensities of the bands in the control and analyzed lanes were compared. Complete hydrolysis of the substrate was taken as 100%. Densitometric analysis was performed using Image Quant 5.2 software (GE Healthcare, Chicago, IL, USA). The specific ODN-hydrolyzing activity of IgG samples was calculated using the following formula: Specific activity = (% of hydrolysis × [ODN, M])/(100 × Reaction time, min × [IgG, mg/mL]).

*2.7. Statistical Analysis*

Statistical analysis was carried out in the STATISTICA 10 (StatSoft. Inc., Tulsa, OK, USA). Experimental results are presented as mean ± standard deviation (SD). The data were evaluated for normality using the Shapiro–Wilk test. Since the data were not normally distributed, the nonparametric two-tailed Mann–Whitney U test was used to analyze the significance of differences in the level of ODN hydrolysis by antibodies of patients and healthy individuals. A *p*-value of less than 0.05 was considered to be statistically significant. Graphs were plotted using Origin 2019 (OriginLab Corporation, Northampton, MA, USA).

**3. Results**

*3.1. Clinical Characteristics of Patients and Healthy Individuals*

The sample for this study included five SLE patients and five healthy controls. All SLE patients were positive for anti-dsDNA and anti-ssDNA antibodies, and healthy individuals were negative. The age of participants in the two groups did not differ significantly (two-tailed Mann–Whitney U Test, *p* = 0.603). All SLE patients were women. The healthy individuals included 3 men and 2 women. The average duration of SLE was 3.4 ± 1.8 years. The mean SELENA-SLEDAI score was 9.2 ± 0.8. Individual information for each participant is presented in Table 2.

**Table 2.** Clinical data of SLE patients and healthy individuals.

| Group | No. of IgG Samples | Sex | Age, Years | Disease Duration, Years | Course of SLE | SELENA-SLEDAI Score |
|---|---|---|---|---|---|---|
| SLE | SLE22 | F | 34 | 4 | subacute | 10 |
| | SLE23 | F | 25 | 5 | subacute | 8 |
| | SLE24 | F | 47 | 5 | subacute | 9 |
| | SLE33 | F | 58 | 2 | chronic | 9 |
| | SLE44 | F | 71 | 1 | subacute | 10 |
| Mean ± SD | - | - | 47.0 ± 18.4 | 3.4 ± 1.8 | - | 9.2 ± 0.8 |
| Healthy subjects | HS74 | F | 61 | - | - | - |
| | HS81 | M | 25 | - | - | - |
| | HS45 | M | 43 | - | - | - |
| | HS20 | F | 42 | - | - | - |
| | HS40 | M | 27 | - | - | - |
| Mean ± SD | - | - | 39.6 ± 14.6 | - | - | - |

Polyclonal IgG samples were purified from the serum of each participant using affinity chromatography. It was previously shown that this method produces pure IgG without any impurities [42,43]. The obtained IgG samples were used for further analysis.

*3.2. Hydrolysis of Immobilized ODNs on Microarray Surface by Antibodies*

The linker length between the oligonucleotide and the surface is important for analyzing ODN hydrolysis on a microarray. If the linker is small, it may reduce hydrolysis efficiency, so we introduced an extra 6 triethylene glycol phosphate residues, in addition to the standard aminohexyl linker, in the synthesis of ODN. Linker extension has previously been shown to improve on-microchip PCR efficiency [45].

Figure 1 shows a view of the microarray with immobilized ODNs (Figure 1A) and the change in fluorescence before and after the application of IgG samples from patients and healthy donors (Figure 1B,C). It was shown that after incubation with antibodies from SLE patients, fluorescence decreases significantly (Figure 1B). In the case of antibodies from healthy donors, the decrease in fluorescence over the same time is less significant (Figure 1C).

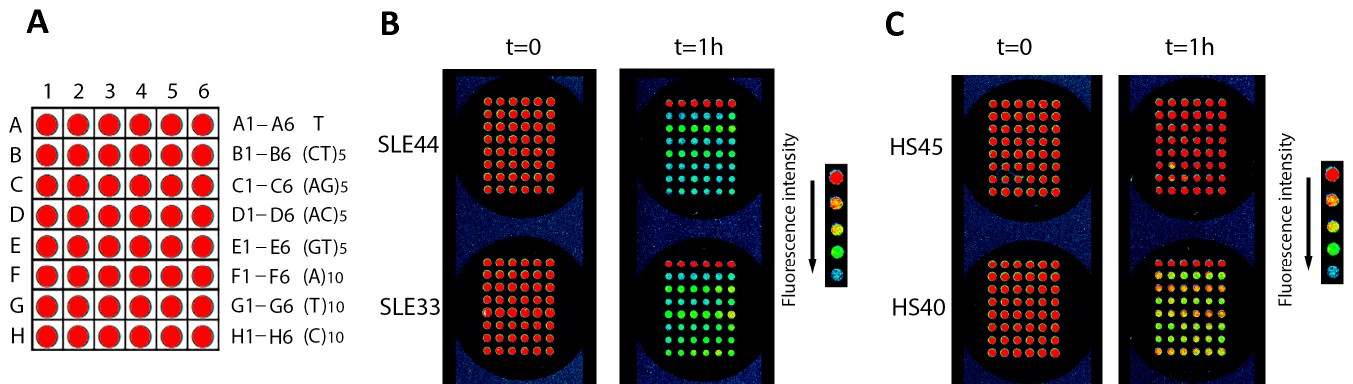

**Figure 1.** View of the microarray and arrangement of ODNs (**A**); change in fluorescence after 1 h incubation with IgG samples of SLE patients (**B**) and healthy subjects. (**C**) A color map representing fluorescence intensity is shown in the legend (**B,C**). IgG sample No. of patients (SLE33 and SLE44) and healthy individuals (HS40 and HS45) is indicated.

The selection of the minimum length of oligonucleotides for the experiments was carried out, based on the results of the efficiency of hydrolysis of ODNs with lengths of 6 and 10 nucleotides (Figure 2). DNase I was used as a positive control for hydrolysis on the surface of the microarray (Figure 2A). Longer ODNs were hydrolyzed more efficiently by DNase I. A similar pattern was observed for IgG sample SLE22 of SLE patient (Figure 2B). It was shown that longer ODNs $(AG)_5$ and $(CT)_5$ are hydrolyzed by antibodies more efficiently than shorter ODNs $(AG)_3$ and $(CT)_3$, respectively. Further extension of the ODN has little effect, so ODNs with a length of 10 nucleotides were chosen for the experiments.

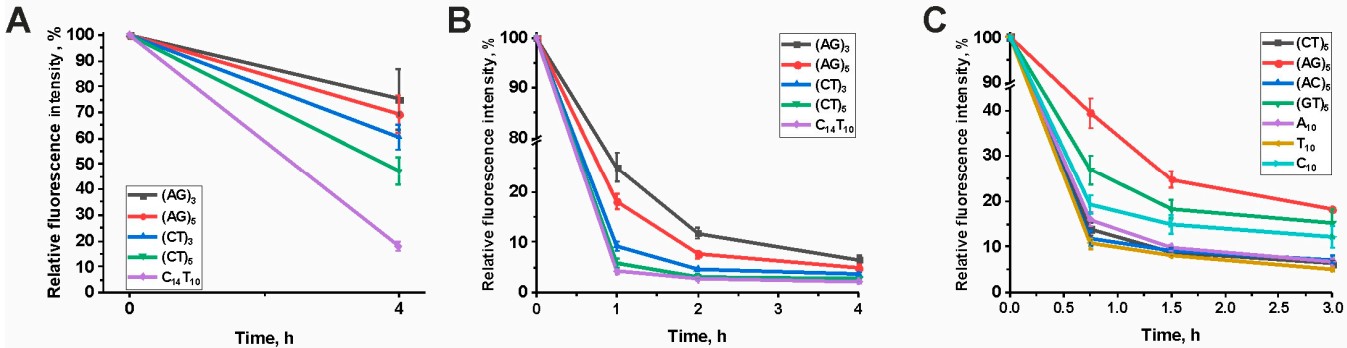

**Figure 2.** Hydrolysis of ODNs of various lengths by DNase I (**A**) and IgG of SLE patients (**B**); time dependence of hydrolysis of ODNs of the same length by IgG of SLE patients. (**C**) Each point represents the mean ± SD of 3–6 replicates.

The reaction kinetics of hydrolysis of model ODNs of the same length by IgG sample SLE22 of SLE patient are presented in Figure 2C. It can be seen that the main hydrolysis occurs in an interval of up to 1.5 h. Model ODNs are hydrolyzed with different efficiencies.

The decrease in the fluorescence level of various ODNs due to hydrolysis by IgG of SLE patients and healthy donors is presented in Figure 3. Data are presented as the ratio of fluorescence after incubation with IgG (F1) and initial fluorescence (F0). Therefore, the lower the F1/F0 ratio, the greater the decrease in fluorescence, and consequently, the higher the level of hydrolysis. It was shown that ODN $(AG)_5$ was hydrolyzed least efficiently by each antibody samples of SLE patients (Figure 3). The hydrolysis efficiency of the remaining ODNs had similar values and varied for different IgG samples. The ODN-hydrolyzing activity of antibodies from healthy donors was lower on average by 2.5–3.5 times (depending on ODN) than in patients with SLE. In the control well with buffer, the fluorescence level for each ODN remained almost unchanged (average of 3 experiments, 18 spots) (Figure 3C).

The data were then recalculated as a percentage of hydrolysis (Figure 4). It was shown that almost all ODNs were hydrolyzed by IgG samples from patients more effectively (approximately 2.6–3.9 times by median values) than healthy donors ($p < 0.022$, Mann–Whitney test). The exception was ODN $(AG)_5$, for which no significant differences in the level of hydrolysis were found (Figure 4).

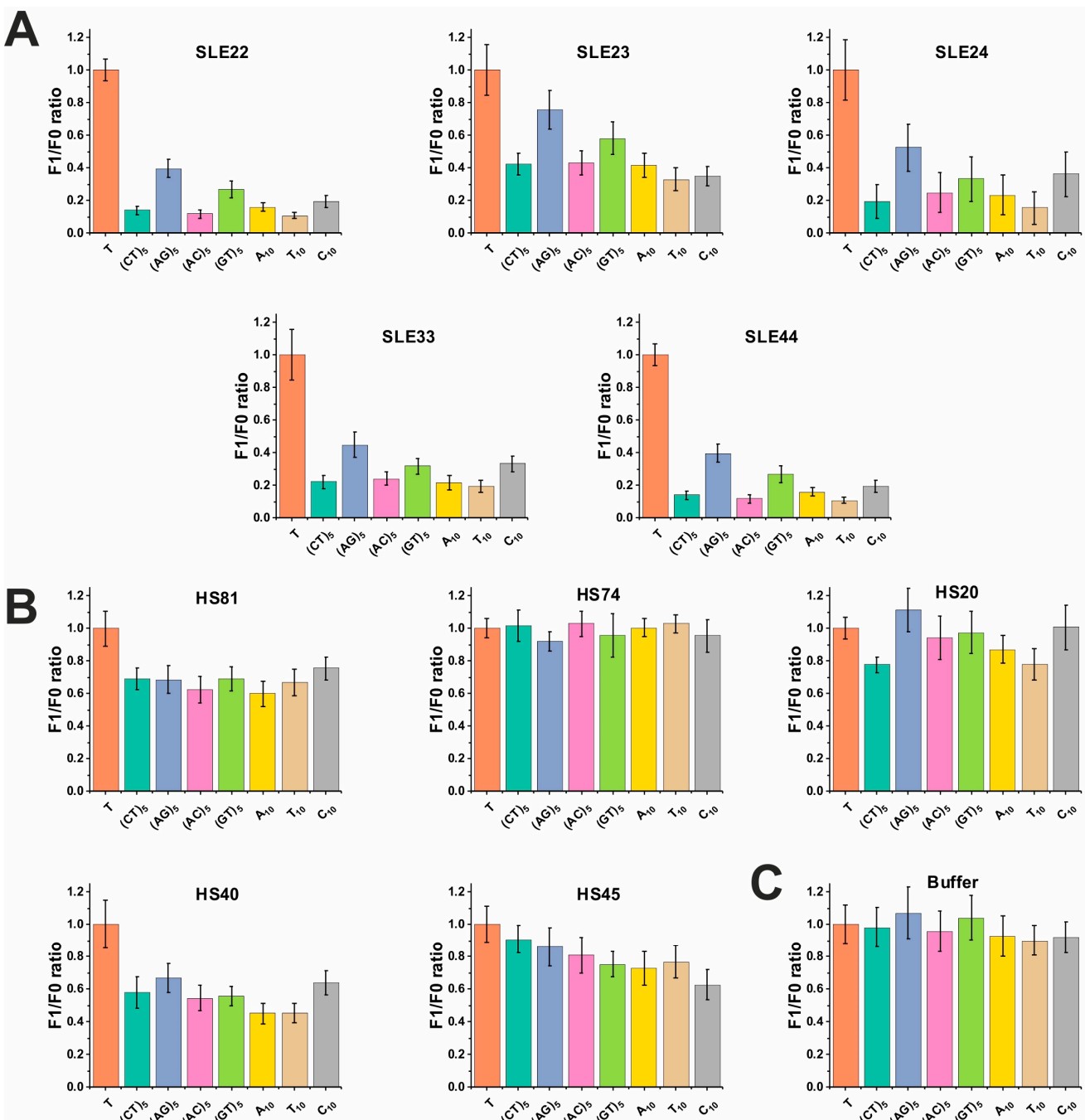

**Figure 3.** Ratio of fluorescence units after 1 h of incubation (F1) to the initial fluorescence (F0) in the presence of IgG antibodies from SLE patients, (**A**) healthy donors, (**B**) and control buffer containing no antibodies. (**C**) Mononucleotide T, which is not hydrolyzed, was used to normalize fluorescence. The data are presented as the mean ± SD of 3 replicates for each bar.

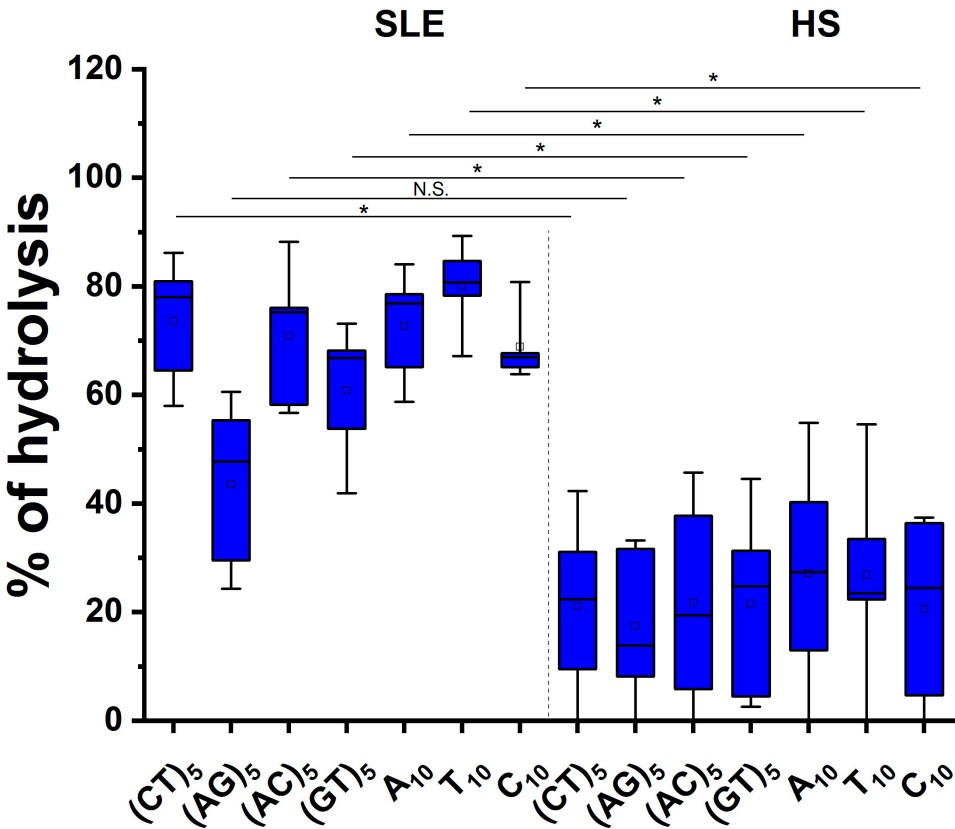

**Figure 4.** Comparison of the level of relative hydrolysis of ODNs on the microarray surface after 1 h incubation with IgG antibodies of SLE patients (n = 5) and healthy donors (n = 5). Complete hydrolysis of ODNs is taken as 100%. The boxplots show the median values (horizontal line), the mean values (empty square), the lower and upper quantiles (colored boxes), and the min and max (error bar). *, significantly different ($p < 0.02$, Mann–Whitney test) compared to the control group; N.S.—not significant.

### 3.3. Hydrolysis of ODN by Antibodies in Solution

An example of an electrophoretic analysis of ODN hydrolysis in solution by DNase I and the IgG sample SLE33 of SLE patient is presented in Figure 5. ODNs have been shown to be hydrolyzed with varying efficiencies by antibodies and DNase I. Hydrolysis of ODNs by IgG antibodies, in contrast to DNase I, led to the formation of many product variants. Interestingly, some hydrolysis products migrate more slowly than intact ODNs (Figure 5). The abnormal electrophoretic mobility of the hydrolysis products is explained by the presence of a positive charge on the ROX dye, which partially neutralizes the negative charge of ODN. This was confirmed by an additional experiment (Supplementary Figure S1). This experiment also showed that the end product of hydrolysis is predominantly a mononucleotide and IgG-dependent fragmentation of the decanucleotide leads to the formation of four hydrolysis products (as also seen in Figure 5B). Analysis of the specificity of ODN hydrolysis by DNase I showed that ODN $(AT)_5$ was hydrolyzed with the greatest efficiency (Figure 5A). The remaining ODNs were hydrolyzed less efficiently. Hydrolysis of $(AT)_5$ can be explained by the formation of duplexes in solution and also by the fact that DNase I is known to be specific for double-stranded DNA molecules, but also hydrolyzes single-stranded DNA. The data obtained were consistent with the literature on the specificity of DNase I [46]. Differences in the patterns of hydrolysis of ODN by DNase I and IgG sample SLE33 prove the absence of DNase I impurities in the analyzed IgG.

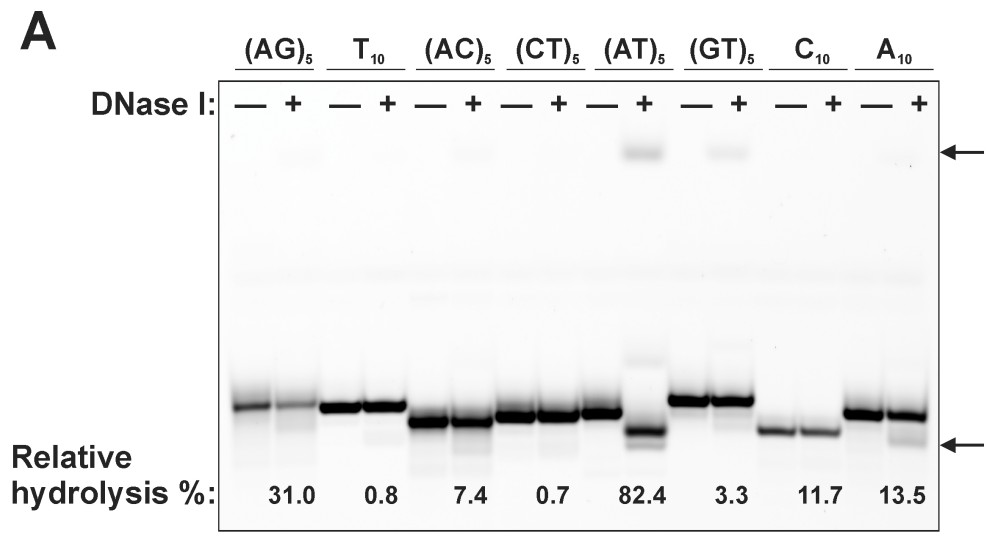

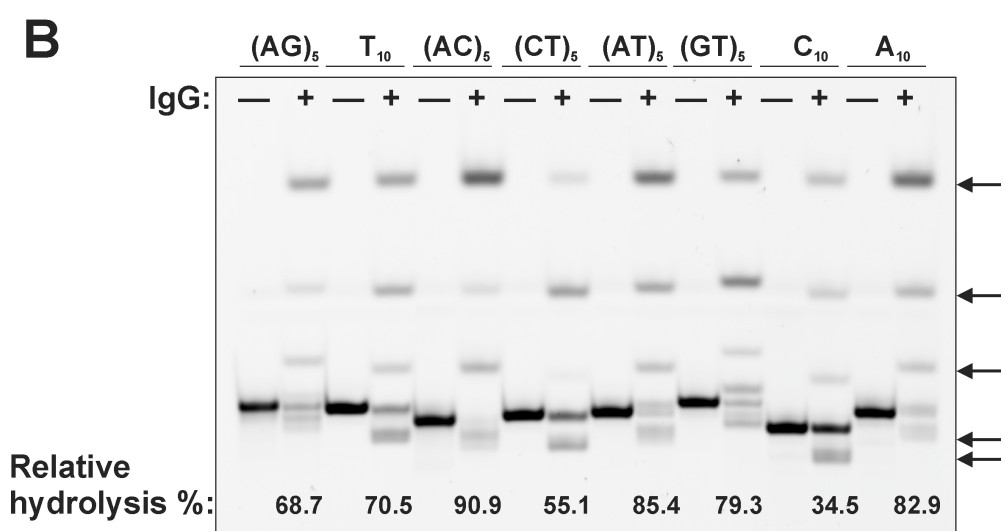

**Figure 5.** Analysis of the relative hydrolysis level of ODN in solution by DNase I (**A**) and the IgG sample SLE33 (**B**) using electrophoresis. The incubation time for ODNs with IgG sample SLE33 (0.67 μM) and DNase I (0.1 U/μL) was 15 min and 30 min, respectively. Complete hydrolysis of ODNs is taken as 100%. Arrows indicate hydrolysis products.

The relative percentage of hydrolysis of individual IgG samples from patients and healthy individuals is presented in Figure 6. Variation in the efficiency of ODN hydrolysis by different IgGs has been shown. ODN $(AC)_5$ was hydrolyzed most efficiently by antibodies of SLE patients. ODN $(C)_{10}$ was hydrolyzed the least in each experiment. The remaining ODNs were hydrolyzed with intermediate efficiency. Low activity of antibody samples (about 6–16 times) was observed in the case of healthy donors compared to SLE patients.

Relative DNA-hydrolyzing activity of IgG antibodies was converted to specific activity. The level of specific activity of antibody samples of patients with SLE was significantly 6–16 times higher ($p < 0.012$, Mann–Whitney test) than in healthy donors (Figure 7). The data on the high level of DNA-hydrolyzing activity of antibodies of SLE patients in solution were consistent with the data on relative hydrolysis of ODNs on the microarray surface (Figure 4). However, the efficiency of hydrolysis of individual ODNs on the surface of the microarray and in solution differed.

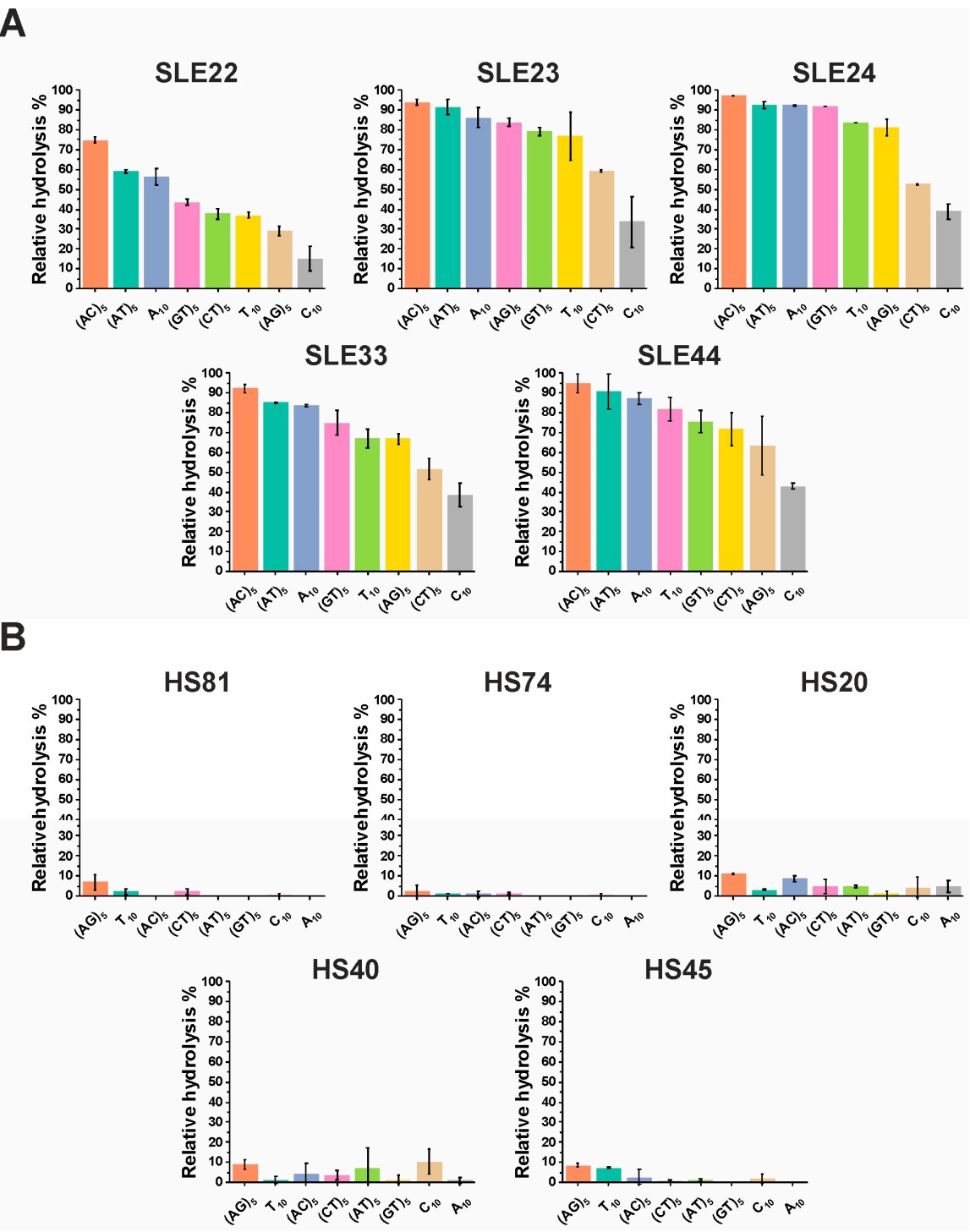

**Figure 6.** The level of relative hydrolysis of ODNs in solution by IgG samples from patients with SLE (**A**) and healthy donors. (**B**) The incubation time of ODNs with IgG samples was 15 min for SLE33 and SLE22 and 30 min for other samples. Complete hydrolysis of ODNs is taken as 100%. The data are presented as the mean ± SD of 2 replicates for each bar.

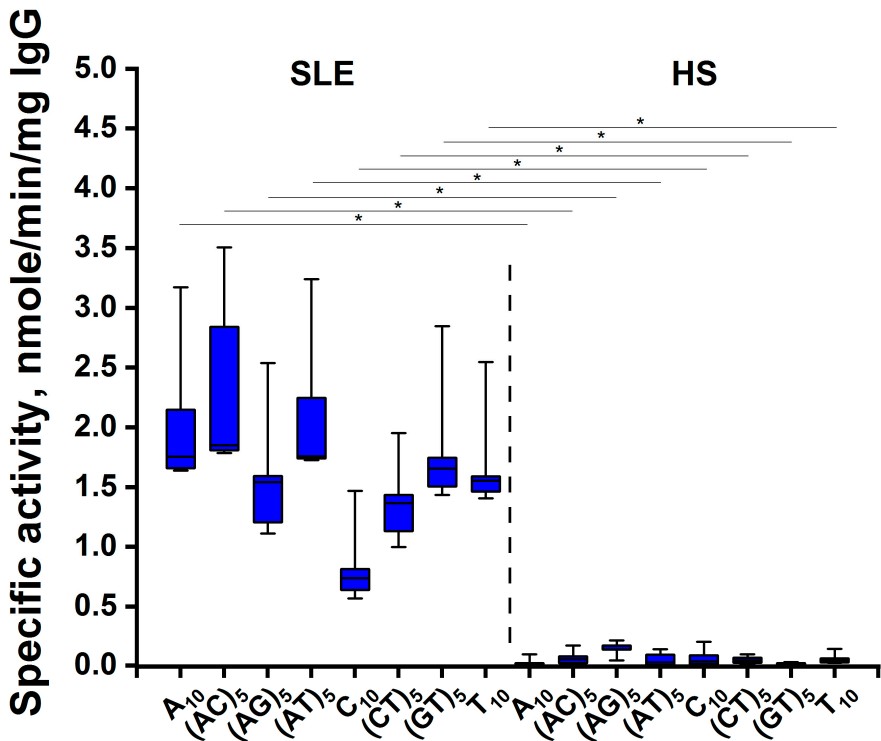

**Figure 7.** Comparison of the level of specific ODN-hydrolyzing activity of IgG samples from SLE patients (n = 5) and healthy donors (n = 5) in solution. The boxplots show the median values (horizontal line), the lower and upper quantiles (colored boxes), and the min and max (error bar). *, significantly different ($p < 0.012$, Mann–Whitney test) compared to control group.

The obtained data on the efficiency of ODN hydrolysis by antibodies in solution and on the microarray surface are summarized in Figure 8. These data indicate significant differences in the efficiency and specificity of ODN hydrolysis on the microarray surface and in solution. For example, ODN $(AG)_5$ was poorly hydrolyzed on the microarray surface, but ODN $C_{10}$ was less hydrolyzed in solution. In addition, differences were identified depending on the specific IgG sample.

**A**      **Hydrolysis of ODN by antibodies on the microchip surface**

SLE33 : $T_{10} \geq A_{10} \geq (CT)_5 \geq (AC)_5 \geq (GT)_5 \geq C_{10} \geq (AG)_5$

SLE44 : $T_{10} > C_{10} \geq A_{10} = (CT)_5 \geq (AC)_5 \geq (GT)_5 > (AG)_5$

SLE22 : $T_{10} \geq (AC)_5 > (CT)_5 > A_{10} > C_{10} > (GT)_5 > (AG)_5$

SLE23 : $T_{10} \geq C_{10} \geq A_{10} \geq (CT)_5 \geq (AC)_5 \geq (GT)_5 \geq (AG)_5$

SLE24 : $T_{10} > (CT)_5 > A_{10} \geq (AC)_5 > (GT)_5 \geq C_{10} > (AG)_5$

**B**      **Hydrolysis of ODN by antibodies in solution**

SLE33 : $(AC)_5 > (AT)_5 > A_{10} > (GT)_5 \geq T_{10} \geq (AG)_5 > (CT)_5 > C_{10}$

SLE44 : $(AC)_5 \geq (AT)_5 \geq A_{10} \geq T_{10} \geq (GT)_5 \geq (CT)_5 \geq (AG)_5 > C_{10}$

SLE22 : $(AC)_5 > (AT)_5 \geq A_{10} > (GT)_5 > (CT)_5 \geq T_{10} > (AG)_5 > C_{10}$

SLE23 : $(AC)_5 \geq (AT)_5 \geq A_{10} \geq (AG)_5 \geq (GT)_5 \geq T_{10} > (CT)_5 > C_{10}$

SLE24 : $(AC)_5 > (AT)_5 \geq A_{10} \geq (GT)_5 > T_{10} \geq (AG)_5 > (CT)_5 > C_{10}$

**Figure 8.** Efficiency of ODN hydrolysis on the microchip surface (**A**) and in solution (**B**) by IgG samples of SLE patients in decreasing order.

## 4. Discussion

The interaction between natural catalytic antibodies and DNA can be considered as an antigen–antibody and enzyme–substrate interaction. The data obtained indicate differences in the recognition and hydrolysis of ODNs on the microarray surface and in solution by anti-DNA antibodies of SLE patients. It is impossible to directly compare the efficiency of ODN hydrolysis in solution and on the surface, because it is difficult to achieve exactly the same reaction conditions. However, the results obtained indicate a lower efficiency of ODN hydrolysis on the surface than in solution (Figures 3, 4 and 6A). For example, in the case of hydrolysis of ODN $A_{10}$ by IgG sample SLE33, the relative percentage of hydrolysis on the microarray surface and in solution was about the same (78.6 ± 4.3% and 83.5 ± 0.7%, respectively), but the incubation time was 4 times less in solution than on the surface. The concentration of IgG was the same (0.67 μM), but the ODN content in the solution and on the surface differed due to the peculiarities of the methods used. More precisely, the concentration of ODN in the solution was 5.7 μM, but on the surface, only 2 μM of ODN was covalently tethered to 0.1 mm$^2$ of the microarray surface. Nevertheless, the results obtained on the presumed lower efficiency of ODN hydrolysis by antibodies on the microarray surface and in solution are consistent with literature data regarding enzymes [37,47–49]. For example, Corn and colleagues showed that the reaction rate constant ($k_{cat}$) for surface hydrolysis of RNA in RNA−DNA heteroduplexes by RNase H on a DNA microarray was approximately 10 times slower than that observed in solution [49]. Although there are data on comparable rates of hydrolysis in solution and on the surface or inverse relationships, the available data are typical for other enzymes, not nucleases [50,51]. The observed differences are associated with the limited amount of surface-tethered substrate (S) and an excess of dissolved enzyme (E), with much of the substrate present as an immobilized enzyme−substrate complex (ES). Moreover, the concentration of immobilized ES is not constant over time, but tends to zero as the reaction is completed [37,47,52]. The conditions of classical enzyme kinetics imply an excess of substrate ([S]>>[E]). Lateral diffusion of the enzyme also contributes to the efficiency of adsorbed substrate hydrolysis [53]. In general, enzymatic reactions involving surface-immobilized substrate differ in kinetics, thermodynamics, and chemical selectivity from reactions in solution [37,47]. The enzymatic reaction with surface-tethered substrate can be described in terms of classical Langmuir adsorption model and Michaelis–Menten kinetics [52,54,55].

The results of this work also indicate differences in the specificity of anti-DNA antibodies to certain ODNs (Figures 3, 4 and 6–8). For example, ODN $T_{10}$ hydrolyzed best on the microarray surface, while ODN $(AC)_5$ hydrolyzed most efficiently in solution (Figure 8). The observed differences in specificity may be related to the different recognition of ODNs on the surface and in solution. It may also be associated with the limited accuracy of the methods used. Nevertheless, the data obtained indicate the formation of antibodies to specific DNA motifs in SLE. Among these antibodies, there may be both high-affinity and catalytic antibodies with lower affinity. Catalytic antibodies may also include antibodies to certain DNA motifs. There is evidence that fragments of single-stranded DNA recognized by anti-DNA antibodies are rich in the following sequences: CACC, CACCC, ACCC, CCCC. Antibodies to such fragments also well recognized the following motifs of dsDNA: 5′-GCG-3′/3′-CGC-5′ [22]. According to our data, IgG of SLE patients efficiently hydrolyze ODN $(AC)_5$ in solution (Figure 8), which may be related to the formation of antibodies to such motifs and their efficient recognition. There is also evidence that recognition and high-affinity DNA binding by the antibody depend on monogamous bivalence, in which both Fab sites of the IgG molecule contact the same polynucleotide chain [20,56]. We used short ODNs (10 nucleotides) to exclude the influence of the second Fab site of an IgG molecule. According to literature data Fab sites of the light or heavy chain of the antibody interact and form tight contacts with 2–4 nucleotides of ODN [57]. Therefore, even short nucleotides are sufficient to be recognized by antibodies. Longer ODNs should be used in further studies to evaluate the contribution of monogamous bivalence to antibody binding to DNA.

This work also shows that the level of ODN-hydrolyzing activity of IgG samples of SLE patients is significantly higher than that of healthy donors (Figures 4 and 7). These results are in agreement with previously obtained data [27,30]. However, previous studies used plasmid DNA (dsDNA) as a substrate. In this work, the ability of anti-DNA antibodies in SLE to hydrolyze short ssDNA was demonstrated. Interestingly, all patients received corticosteroids and other drugs that reduce the level of antinuclear and anti-DNA antibodies and promote B-cell depletion [58–60]. Nevertheless, the work revealed an increase in the level of ODN-hydrolyzing activity of antibodies in SLE compared to healthy individuals. The effect of therapy on the catalytic activity of anti-DNA antibodies in SLE requires further investigation.

Fragmentation of extracellular genomic DNA by blood nucleases leads to the formation of fragments with a periodicity of 10 nucleotides, which is associated with limited access of nucleases to DNA due to the nucleosomal structure [61]. Our work revealed efficient hydrolysis of short 10-nucleotide ODNs by catalytic IgG. Therefore, it can be suggested that catalytic anti-DNA antibodies in SLE, along with blood nucleases, including DNase I, DNase I-like 3, and others [61], are involved in the clearance of extracellular DNA. High-affinity noncatalytic anti-DNA antibodies are also involved in the clearance of extracellular DNA, but they form immune complexes that trigger lupus nephritis [62,63]. Therefore, catalytic anti-DNA antibodies that bind DNA with lower affinity may have less pathologic effects than classical anti-DNA antibodies. Some studies indicate that the concentration of extracellular DNA is positively correlated with the level of anti-nucleosomal or anti-DNA antibodies and SLEDAI scores [64,65]. Therefore, it can be suggested that anti-DNA antibodies are inducibly increased in response to elevated levels of extracellular DNA in SLE. In addition, there is evidence that extracellular DNA can be adsorbed on the surface of blood cells and circulate for a long time [66]. Such cell-surface-bound extracellular DNA was also found in rheumatic diseases [67]. Therefore, the hydrolysis of surface-bound DNA by anti-DNA antibodies in SLE identified in this work may play an important role in-vivo. Altogether, catalytic anti-DNA antibodies along with blood DNAases may be considered an important component of the clearance system against circulating extracellular DNA in SLE and other diseases.

## 5. Conclusions

In this work, differences in efficiency and specificity of the hydrolysis of ODN on the microarray surface and in solution by anti-DNA antibodies of SLE patients were identified. The data obtained indicate more efficient hydrolysis of dissolved than surface-tethered ODNs. The identified differences in the recognition and hydrolysis of ODNs may be associated with the formation of antibodies to specific DNA motifs in SLE. The difference in recognition and hydrolysis of DNA in solution and on the surface needs to be considered both in DNA microarray applications and to understand the role of anti-DNA antibodies in vivo.

**Supplementary Materials:** The following supporting information can be downloaded at: https://www.mdpi.com/article/10.3390/cimb45120617/s1, Figure S1: Analysis of the effect of ROX dye on the electrophoretic mobility of ODNs and identification of the final product of hydrolysis by antibodies of SLE patients.

**Author Contributions:** Conceptualization, E.A.E., E.V.K., A.N.S., G.A.N. and V.N.B.; methodology, E.A.E., E.V.K., A.N.S., G.A.N. and V.N.B.; software, E.A.E. and E.V.K.; validation, T.S.N., A.N.S., A.E.S., G.A.N. and V.N.B.; formal analysis, E.A.E. and E.V.K.; investigation, T.S.N., E.A.E. and E.V.K.; resources, A.N.S., A.E.S., G.A.N. and V.N.B.; data curation, A.N.S. and G.A.N.; writing—original draft preparation, T.S.N., E.A.E. and E.V.K.; writing—review and editing, E.A.E., A.N.S., A.E.S., G.A.N. and V.N.B.; visualization, E.A.E.; supervision, G.A.N. and V.N.B.; project administration, V.N.B.; funding acquisition, V.N.B. All authors have read and agreed to the published version of the manuscript.

**Funding:** This work was funded by the Russian Science Foundation [grant number 23-15-00357 (the main part of the work on anti-DNA antibody analysis)] and the Russian state-funded project for

ICBFM SB RAS [grant numbers 121031300042-1 (part of the work on ODN synthesis and microarray printing) and 121031300041-4 (part of the research on DNase I)].

**Institutional Review Board Statement:** The study was conducted in accordance with the Declaration of Helsinki and approved by the Local Ethics Committee of the Institute of Chemical Biology and Fundamental Medicine (protocol N3 from 19 June 2023).

**Informed Consent Statement:** Informed consent was obtained from all subjects involved in the study.

**Data Availability Statement:** Data supporting the reported results are presented in the manuscript. Additional and raw data are available upon request from the corresponding author.

**Acknowledgments:** The authors would like to thank all patients and healthy subjects who participated in the study. The authors also thank Mark M. Melamud for assistance with the purification of IgG samples.

**Conflicts of Interest:** The authors declare no conflict of interest.

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
