# Peer review of "Hydrolysis of Oligodeoxyribonucleotides on the Microarray Surface and in Solution by Catalytic Anti-DNA Antibodies in Systemic Lupus Erythematosus"

_cimb, doi:10.3390/cimb45120617_

Round 1
Reviewer 1 Report
Comments and Suggestions for Authors
In this article, Novikova et al analyzed the hydrolysis of short oligodeoxyribonucleotides (ODNs) immobilized on the microarray surface and in solution by catalytic anti-DNA antibodies of SLE patients. It has been shown that IgG antibodies from SLE patients hydrolyze ODNs more effectively both in solution and on the surface compared to IgG from healthy individuals. The data obtained indicate a more efficient hydrolysis of ODNs in solution than immobilized ODNs on the surface. In addition, differences in the specificity of recognition and hydrolysis of certain ODNs by anti-DNA antibodies were revealed, which indicates the formation of autoantibodies to specific DNA motifs in SLE. The findings are very interesting. However, I have some questions as follows.
major concerns)
1) In this article, anti-DNA antibodies taken from the same patient have different degradation speeds depending on the type of ODN. What are the possible clinical implications of the different degradation speeds of different types of ODNs?
2) In this article, the speed of degradation of the same ODN varies from patient to patient. In this regard, as well, is there any correlation with clinical practice, such as patients with faster degradation speed of ODNs are less likely to have worse disease? Since the number of patients is small, even if there is no significant difference, any trend would be helpful for future research, so please provide this information.
3) What is the possible significance of anti-DNA antibodies, which facilitate DNA degradation? Please mention in the discussion section.
4) According to the present data, anti-DNA antibodies in SLE help in the degradation of DNA. When considering the pathogenesis of SLE, are anti-DNA antibodies considered a cause or a result? This would be a very important consideration for the pathogenesis of SLE. Please add in the discussion section.
Author Response
Dear Reviewer,
We thank the reviewer for the positive evaluation of our study and valuable suggestions. Your suggestions were helpful especially for the discussion section.
Below we answer your suggestions point by point. Your comments are in italics.
In this article, Novikova et al analyzed the hydrolysis of short oligodeoxyribonucleotides (ODNs) immobilized on the microarray surface and in solution by catalytic anti-DNA antibodies of SLE patients. It has been shown that IgG antibodies from SLE patients hydrolyze ODNs more effectively both in solution and on the surface compared to IgG from healthy individuals. The data obtained indicate a more efficient hydrolysis of ODNs in solution than immobilized ODNs on the surface. In addition, differences in the specificity of recognition and hydrolysis of certain ODNs by anti-DNA antibodies were revealed, which indicates the formation of autoantibodies to specific DNA motifs in SLE. The findings are very interesting. However, I have some questions as follows.
Reply: Thank you for your thorough analysis and high appreciation of the manuscript.
major concerns)
1) In this article, anti-DNA antibodies taken from the same patient have different degradation speeds depending on the type of ODN. What are the possible clinical implications of the different degradation speeds of different types of ODNs?
Reply: Thank you for this question. The main function of anti-DNA antibodies in SLE is to bind/degrade and excrete cell-free DNA/ODN. Therefore, due to the different degradation rates of different types of DNA/ODN by antibodies, some DNA/ODN variants with characteristic sequences may accumulate in higher quantities than others. Some DNA motifs (especially CpG motifs) have high immunostimulatory potential. Therefore, the accumulation of certain DNA/ODNs may either enhance or reduce the inflammatory response, depending on the immunostimulatory potential of these accumulating DNA/ODNs. Cell-free DNA levels have been reported to be elevated in patients with SLE compared to healthy controls [1, 2]. However, the immunostimulatory potential of cell-free DNA in SLE is still poorly understood. It can be speculated that antibodies with DNA-hydrolyzing properties contribute to the diversity of cell-free DNA fragments in SLE. The clinical role of catalytic anti-DNA antibodies in SLE remains to be elucidated in future experiments.
2) In this article, the speed of degradation of the same ODN varies from patient to patient. In this regard, as well, is there any correlation with clinical practice, such as patients with faster degradation speed of ODNs are less likely to have worse disease? Since the number of patients is small, even if there is no significant difference, any trend would be helpful for future research, so please provide this information.
Reply: Indeed, it is important to identify the association of ODN degradation rate with clinical parameters. We performed correlation analysis, but unfortunately we did not find any significant associations. There were no associations between high level/low ODN hydrolysis and clinical parameters (neither quantitative nor qualitative). This is due to the small sample size (5 SLE patients and 5 healthy subjects) and the small variation in clinical variables (e.g., the SELENA-SLEDAI index ranged from 8 to 10 (please see Table 2)). We plan to conduct further experiments on larger samples to identify associations with clinic. So thank you for this suggestion.
3) What is the possible significance of anti-DNA antibodies, which facilitate DNA degradation? Please mention in the discussion section.
Reply: Thank you for this question. According to numerous studies, cell-free DNA levels are elevated in SLE [1, 2]. Cell-free DNA/ODN may have high immunostimulatory potential. Various blood nucleases including DNase I, DNase I-like 3 and others are involved in the degradation and excretion of cell-free DNA [3]. It is conceivable that catalytic anti-DNA antibodies in SLE, along with blood nucleases, are involved in the clearance of cell-free DNA. High-affinity noncatalytic anti-DNA antibodies are also involved in clearance of extracellular DNA, but they form immune complexes that trigger lupus nephritis. Therefore, catalytic anti-DNA antibodies that bind DNA with lower affinity may have less pathologic effects than classical anti-DNA antibodies. We have added this information to the manuscript (please see lines 431-435).
4) According to the present data, anti-DNA antibodies in SLE help in the degradation of DNA. When considering the pathogenesis of SLE, are anti-DNA antibodies considered a cause or a result? This would be a very important consideration for the pathogenesis of SLE. Please add in the discussion section.
Reply: Thank you for this suggestion. Some studies indicate that the concentration of extracellular DNA is positively correlated with the level of anti-nucleosomal or anti-DNA antibodies and SLEDAI scores [4, 5]. Therefore, it can be suggested that anti-DNA antibodies are inducibly increased in response to elevated levels of extracellular DNA in SLE. We have added this information to the manuscript (please see lines 439-442).
References:
- Xu, Y.; Song, Y.; Chang, J.; Zhou, X.; Qi, Q.; Tian, X.; Li, M.; Zeng, X.; Xu, M.; Zhang, W.; et al. High Levels of Circulating Cell‐free DNA Are a Biomarker of Active SLE. Eur J Clin Investigation 2018, 48, e13015, doi:10.1111/eci.13015.
- Duvvuri, B.; Lood, C. Cell-Free DNA as a Biomarker in Autoimmune Rheumatic Diseases. Front. Immunol. 2019, 10, 502, doi:10.3389/fimmu.2019.00502.
- Han, D.S.C.; Ni, M.; Chan, R.W.Y.; Chan, V.W.H.; Lui, K.O.; Chiu, R.W.K.; Lo, Y.M.D. The Biology of Cell-Free DNA Fragmentation and the Roles of DNASE1, DNASE1L3, and DFFB. The American Journal of Human Genetics 2020, 106, 202–214, doi:10.1016/j.ajhg.2020.01.008.
- Abdelal, I.T.; Zakaria, M.A.; Sharaf, D.M.; Elakad, G.M. Levels of Plasma Cell-Free DNA and Its Correlation with Disease Activity in Rheumatoid Arthritis and Systemic Lupus Erythematosus Patients. The Egyptian Rheumatologist 2016, 38, 295–300, doi:10.1016/j.ejr.2016.06.005.
- Hendy, O.M.; Motalib, T.A.; El Shafie, M.A.; Khalaf, F.A.; Kotb, S.E.; Khalil, A.; Ali, S.R. Circulating Cell Free DNA as a Predictor of Systemic Lupus Erythematosus Severity and Monitoring of Therapy. Egyptian Journal of Medical Human Genetics 2016, 17, 79–85, doi:10.1016/j.ejmhg.2015.07.001.
Best regards
Authors
Reviewer 2 Report
Comments and Suggestions for Authors
The authors investigated the hydrolysis of short ODNs that were immobilized on a microarray surface or present in solution, using catalytic anti-DNA antibodies from SLE patients. They reported that antibodies from SLE patients hydrolyze ODNs more effectively both in solution and on the surface compared to those from healthy individuals. A more effective hydrolysis of ODNs in solution by SLE antibodies was detected as compared to those immobilized on the microarray surface by IgG from SLE patients. They suggested that differences in recognition and hydrolysis of surface-bound and solubilized ODNs by anti-DNA autoantibodies need to be considered in DNA microarray applications. This article presents an interesting finding, but some fundamental issues need more clarity or discussion before considering the publication of this work.
1. What theoretical or clinical characteristics of anti-DNA antibodies is the basis for designing ODNs? This issue deserves further elaboration as it is relevant to whether there is a certain degree of clinical similarity.
2. The SLE patients recruited in this study received at least two types of drugs that affect the immune system. Do these drugs affect the property/generation of autoantibodies? The influence and difference of these drugs on development of autoantibody profiles merits further discussion.
3. As known, anti-DNA antibodies may recognize single-stranded (ssDNA), double-stranded (dsDNA) and specific forms of DNA? Did the authors separately isolate these antibodies with different binding properties and perform subsequent analysis? This can eliminate the mutual interference of anti-DNA antibodies with different characteristics.
4. Evidence indicates that some ANAs bind DNA or associated nucleosome proteins, whereas other ANAs bind protein components of complexes of RNA and RNA-binding proteins. Does the ODNs used in this study take these factors into account? It would be important to clarify this concern.
Author Response
Dear Reviewer,
The authors deeply appreciate your thorough analysis of our manuscript.
Below we answer your suggestions point by point. Your comments are in italics.
The authors investigated the hydrolysis of short ODNs that were immobilized on a microarray surface or present in solution, using catalytic anti-DNA antibodies from SLE patients. They reported that antibodies from SLE patients hydrolyze ODNs more effectively both in solution and on the surface compared to those from healthy individuals. A more effective hydrolysis of ODNs in solution by SLE antibodies was detected as compared to those immobilized on the microarray surface by IgG from SLE patients. They suggested that differences in recognition and hydrolysis of surface-bound and solubilized ODNs by anti-DNA autoantibodies need to be considered in DNA microarray applications. This article presents an interesting finding, but some fundamental issues need more clarity or discussion before considering the publication of this work.
Reply: Thank you for your thoughtful analysis and evaluation of our manuscript.
- What theoretical or clinical characteristics of anti-DNA antibodies is the basis for designing ODNs? This issue deserves further elaboration as it is relevant to whether there is a certain degree of clinical similarity.
Reply: Thank you for this question. For this work, we developed model ODNs. First, we included the ODNs consisting entirely of adenosines, cytidines, and thymidines (A10, C10, T10). It is impossible to synthesize ODNs composed entirely of guanosines due to the formation of conglomerates during synthesis. Second, we included ODNs with alternating nucleotides [(AC)5, (AT)5, (GT)5, (AG)5, (CT)5]. In the third, we based on data on specific single-stranded DNA motifs recognized by anti-DNA antibodies. In particular, Wang et al. showed that fragments of single-stranded DNA recognized by anti-DNA antibodies are rich in the following sequences: CACC, CACCC, ACCC, CCCC [1]. In addition, we included ODNs of different lengths but similar composition [(CT)3 and (CT)5; (GT)3 and (GT)5; and the long ODN C14T10]. We have added this information to the manuscript (please see lines 143-146). In the future, we plan to expand the patient sample and the list of ODNs by adding new ODNs of a different composition.
- The SLE patients recruited in this study received at least two types of drugs that affect the immune system. Do these drugs affect the property/generation of autoantibodies? The influence and difference of these drugs on development of autoantibody profiles merits further discussion.
Reply: Thank you for this suggestion. Indeed, each recruited patient received at least two types of drugs that affect the immune system, one of which belonged to the corticosteroid class (dexamethasone, prednisolone, methylprednisolone, or betamethasone). Other drugs included methotrexate, hydroxychloroquine, celecoxib, azathioprine, filgrastim, tenoxicam, and mycophenolate mofetil. Corticosteroids are known to significantly reduce autoantibody production, including anti-DNA antibodies [2, 3]. Other drugs such as methotrexate deplete B-cells and reduce the production of antinuclear and anti-DNA antibodies [4]. Therefore, treatment of the SLE patients included in the study may have affected the levels of anti-DNA antibodies. No data on the effect of these drugs on catalytic activity were found. However, it cannot be excluded that these drugs influenced the DNA-hydrolyzing activity of antibodies. Nevertheless, all patients were treated (the group was homogeneous by type of treatment). In spite of this, this work showed a higher level of ODN-hydrolyzing activity in patients compared to healthy individuals. We have added information about the effect of treatment on the level of anti-DNA antibodies to the manuscript (please see lines 423-428).
- As known, anti-DNA antibodies may recognize single-stranded (ssDNA), double-stranded (dsDNA) and specific forms of DNA? Did the authors separately isolate these antibodies with different binding properties and perform subsequent analysis? This can eliminate the mutual interference of anti-DNA antibodies with different characteristics.
Reply: Thank you for this question. Indeed, anti-DNA antibodies can recognize different forms of DNA. Recognition and high-affinity binding of DNA by the antibody depends on monogamous bivalence, in which both Fab sites of the IgG molecule contact the same polynucleotide chain [5]. We used short ODNs (10 nucleotides) to exclude the influence of the second Fab-site of the IgG molecule. According to the literature, Fab sites of the light or heavy chain of the antibody interact and form tight contacts with 2-4 nucleotides of the ODN [6]. Hence, even short nucleotides are sufficient for recognition by antibodies. This information is included in the discussion section (please see lines 411-416). In this work, we did not isolate antibodies that specifically recognize ssDNA, dsDNA, and other forms of DNA because our goal was to examine the entire pool of anti-DNA antibodies. In the future, we plan to perform antibody fractionation on a DNA-Sepharose column. So thanks for the idea for future studies.
- Evidence indicates that some ANAs bind DNA or associated nucleosome proteins, whereas other ANAs bind protein components of complexes of RNA and RNA-binding proteins. Does the ODNs used in this study take these factors into account? It would be important to clarify this concern.
Reply: Thank you for this suggestion. The ODNs in this study did not contain nucleic acid-binding proteins. Therefore, we can assume that antibodies that bind mainly to nucleic acids were involved in the hydrolysis of ODNs. The influence of nucleosomal and other DNA-binding proteins on the hydrolysis of ODNs by antibodies may be a direction for further studies. Regarding the ODNs used in this work, we were guided only by the data from Wang et al. [1], which showed some sequences that are well recognized by anti-DNA antibodies. However, the information on specific sequences to which DNA-binding proteins bind is still insufficient to design specific ODNs.
References:
- Wang, Y.; Mi, J.; Cao, X. Anti-DNA Antibodies Exhibit Different Binding Motif Preferences for Single Stranded or Double Stranded DNA. Immunology Letters 2000, 73, 29–34, doi:10.1016/S0165-2478(00)00194-2.
- Arbuckle, M.R.; James, J.A.; Kohlhase, K.F.; Rubertone, M.V.; Dennis, G.J.; Harley, J.B. Development of Anti‐dsDNA Autoantibodies Prior to Clinical Diagnosis of Systemic Lupus Erythematosus. Scand J Immunol 2001, 54, 211–219, doi:10.1046/j.1365-3083.2001.00959.x.
- Tseng, C.; Buyon, J.P.; Kim, M.; Belmont, H.M.; Mackay, M.; Diamond, B.; Marder, G.; Rosenthal, P.; Haines, K.; Ilie, V.; et al. The Effect of Moderate‐dose Corticosteroids in Preventing Severe Flares in Patients with Serologically Active, but Clinically Stable, Systemic Lupus Erythematosus: Findings of a Prospective, Randomized, Double‐blind, Placebo‐controlled Trial. Arthritis & Rheumatism 2006, 54, 3623–3632, doi:10.1002/art.22198.
- Böhm, I. Decrease of B-Cells and Autoantibodies after Low-Dose Methotrexate. Biomedicine & Pharmacotherapy 2003, 57, 278–281, doi:10.1016/S0753-3322(03)00086-6.
- Pisetsky, D.S.; Garza Reyna, A.; Belina, M.E.; Spencer, D.M. The Interaction of Anti-DNA Antibodies with DNA: Evidence for Unconventional Binding Mechanisms. IJMS 2022, 23, 5227, doi:10.3390/ijms23095227.
- Andreev, S.L.; Buneva, V.N.; Nevinsky, G.A. How Human IgGs against DNA Recognize Oligonucleotides and DNA. J of Molecular Recognition 2016, 29, 596–610, doi:10.1002/jmr.2559.
Best regards
Authors
Round 2
Reviewer 1 Report
Comments and Suggestions for Authors
Authors replied to my comments appropriately. No additional comments.